# Disease Occurrence and Climatic Factors Jointly Structure Pomelo Leaf Fungal Succession in Disturbed Agricultural Ecosystem

**DOI:** 10.3390/microorganisms12061157

**Published:** 2024-06-06

**Authors:** Feng Huang, Jinfeng Ling, Guohua Li, Xiaobing Song, Rui Liu

**Affiliations:** 1Plant Protection Research Institute, Guangdong Academy of Agricultural Sciences, Key Laboratory of Green Prevention and Control on Fruits and Vegetables in South China Ministry of Agriculture and Rural Affairs, Guangdong Provincial Key Laboratory of High Technology for Plant Protection, Guangzhou 510640, China; rm12407@126.com (F.H.);; 2Institute of Fruit Tree Research, Meizhou Academy of Agricultural and Forestry Sciences, Meizhou 514071, China

**Keywords:** fungal succession, plant pathogen, temperature, precipitation, citrus

## Abstract

For perennial plants, newly emerged organs are fresh hot spots for environmental microbes to occupy and assemble to form mature microbial communities. In the microbial community, some commensal fungi can play important roles in microbial succession, thus significantly improving host plant growth and disease resistance. However, their participating patterns in microbial assembly and succession remain largely unknown. In this study, we profiled the fungal community and found a similar fungal succession pattern of spring-emerged leaves from March to October in two pomelo orchards. Specifically, the fungal species, tracked on the old leaves, dominated the spring leaves after emergence and then decreased in relative abundance. This reduction in priority effects on the spring leaves was then followed by an increase in the number of observed species, Shannon and phylogenetic diversity indices, and the pathogen-associated fungal groups. In addition, we found that the temporal fungal succession on the spring leaves highly correlated with the disease occurrence in the orchards and with the temperature and precipitation variation from spring to summer. Of the pathogen-associated fungal groups, an increase in the relative abundance of *Mycosphaerellaceae*, hosting the causal agent of citrus greasy spot, correlated with the occurrence of the disease, while the relative abundance of *Diaporthaceae*, hosting the causal agent of melanose, was extremely low during the fungal succession. These results confirm that the two kinds of pathogen-associated fungal groups share different lifestyles on citrus, and also suggest that the study of temporal fungal succession in microbial communities can add to our understanding of the epidemiology of potential plant pathogens.

## 1. Introduction

The pomelo, along with tangerine, mandarin, and sweet orange, is among the most widely cultivated citrus fruits [1]. China is the largest pomelo producer in the world, accounting for about 53.1% (with 4.95 million tons) of global total pomelo production in 2020 (https://www.tridge.com/intelligences/pomelo/production (accessed on 6 September 2022)). During the past decade, the expansion of the pomelo industry has been impeded by several known citrus fungal diseases in China. Taking Meizhou, one of the largest pomelo production areas in China, as an example, it is intriguing that several destructive fungal diseases have appeared and finally been controlled. Citrus black spot, caused by *Phyllosticta* species, and citrus scab, caused by *Elsinoë* species, were reported in 2010 and 2014, respectively [2,3]. Both diseases are hardly seen; instead, citrus melanose, caused by *Diaporthe* species, and citrus greasy spot, caused by *Zasmidium* species (from the family *Mycosphaerellaceae*), are the major occurring fungal diseases in Meizhou nowadays [4]. The successful control of these fungal diseases mainly involves agricultural management, especially the application of different fungicides [2,3]. Additionally, it can also be associated with the natural dynamics of the fungal pathogens in orchards, which might be explained by the weakening of spread ability and pathogenic severity due to microbial competition in the context of plant holobiont [5,6].

During microbiome assembly and succession, priority effects indicate that the order in which microbial species are introduced to the community affects the species abundance and the community structure and function at a stable state [7,8]. For example, when a citrus shoot sprouts, the newly appeared organ is immediately exposed to at least two types of microbes. Based on previous works, we refer to these two types of microbes as the pathobiota and the defensebiota [9,10]. The pathobiota of a microbiota includes microbial species with potential pathogenic behaviors, and the defensebiota can include microbial species with competitive advantages against the pathobiota [9,11,12]. Studies of synthetic communities indicate that the construction of a simplified community of defensebiota is promising in activating plants to defend against microbial pathogens [13]. However, the pattern of assembly and succession of microbial communities on most agricultural plants is still unclear, which largely impedes the development and application of synthetic communities in improving plant growth and resistance.

Temporal variations in temperature and precipitation are highly influential to the prevalence of plant pathogens by affecting their spread, virulence, and infection rates [14]. The well-known species of citrus pathobiota, *Zasmidium citri*, the causal agent of citrus greasy spot, grows extensively during the summer rainy season [15,16]. Both *Elsinoë fawcettii* and *Diaporthe citri* develop the most severe symptoms at moderate temperatures of about 23 to 28 °C, and also with longer periods of wetness [17]. These studies have shown that the temporal variation in temperature and precipitation contributes to the epidemics of plant pathogens, but how these climatic factors affect the succession of the whole citrus microbiome at the same time remains largely unknown.

In this study, we focused on the pomelo leaves that emerged in the spring of 2021. We followed the growth of these leaves from March to October in two pomelo orchards and profiled their fungal communities based on the method of amplicon sequencing. From the association of the fungal communities with sampling time, we aimed to explain the temporal succession patterns of fungal communities based on their variations in fungal diversity indices and the relative abundances of major groups. By the comparison of the fungal communities on old and new leaves, we wanted to explain to what extent the fungal community on new leaves integrates with the fungi sourced from the fungal community on old leaves. Lastly, with the correlation analysis of plant disease occurrence and climatic factors, we tried to figure out their influences on the fungal succession patterns.

## 2. Materials and Methods

### 2.1. Orchard Information and Disease Investigation

Two pomelo orchards were selected in Meizhou city, which is one of the most important pomelo-producing areas worldwide. The Chengjiang orchard (Chengjiang hereafter, 116°4′29″ E, 24°16′30″ N) was in Chengjiang town of Meixian district, and the orchard size was about 3.3 hm^2^. The Xiyang orchard (Xiyang, 116°15′42″ E, 24°16′21″ N) was in Xiyang town of Meijiang district, and the orchard size was about 1.3 hm^2^. Both orchards were located by the hillside, and the altitude was about 100 and 113 m, respectively. The two orchards were human-disturbed with regular agricultural management and fungicide application (Appendix A). From March to October 2021, pomelo diseases were investigated and pomelo leaves were sampled about twice a month in both orchards, except in mid-April and mid-May due to heavy rains. In March 2021, the disease investigation and leaf sampling were conducted on both old leaves (emerged before 2021) and spring leaves (emerged in the spring of 2021); hereafter, the investigation was only performed on spring leaves from April to October. Several common pomelo diseases (e.g., citrus anthracnose, melanose, greasy spot, brown spot, black spot, scab, and canker) were included in our investigation. However, only the diseases greasy spot and melanose were found during our investigations in the orchards. For the investigation, six blocks were set in each orchard. Each block contained three rows of pomelo trees, which resulted in about 15 and 30 trees per block in Xiyang and Chengjiang, respectively. For each block, five trees were randomly selected, and for each tree, five branches stretched in four directions and the center were selected to count the number of diseased leaves and all leaves. The diseased leaves were defined by the presence of typical symptoms of greasy spot and melanose. The disease severity of each tree was calculated by the ratio of the diseased leaves to all leaves. In addition, the data on temperature and precipitation in the districts were obtained from the Bureau of Meizhou Meteorological Service (http://gd.cma.gov.cn/mzsqxj/mobile/index.html (accessed on 16 March 2022)).

### 2.2. Leaf Sampling

During every investigation, only the leaves of spring shoots (which emerged from February to April) were sampled. For each block, one leaf with no visible symptoms was picked from each of the five randomly selected trees; five leaves were mixed as one sample. The old leaves were sampled once in mid-March; the spring leaves were sampled 12 times from March to October. In total, 156 samples, consisting of 12 samples of old leaves and 144 samples of spring leaves, were collected from both orchards. To avoid contamination, sterile gloves were worn during sampling and sterile leaf bags were used to hold the leaves. Then, the leaf bags were preserved in ice and sent back to the lab immediately. To capture the whole microbiota loaded on the leaves, the leaves were not surface-sterilized and were stored under −80 °C.

### 2.3. DNA Extraction and Amplicon Sequencing

The collected leaves were ground with liquid nitrogen, and then, about 1 g powder was stored for each sample. The total DNA of leaf microbes was extracted following the protocol of the CTAB (cetyltrimethylammonium bromide) method [18]. DNA concentration and purity were monitored on 1% agarose gels, and then, DNA was diluted to 1 ng/µL with sterile water. The partial nucleotide sequence of nuclear ribosomal internal transcribed spacer (ITS rDNA) was amplified by polymerase chain reaction (PCR) with the fungal primer set ITS1-1F (5′-CTTGGTCATTTAGAGGAAGTAA-3′) and ITS2 (5′-GCTGCGTTCTTCATCGATGC-3′) as described [19]. The PCRs were carried out with 15 µL of Phusion^®®^ High-Fidelity PCR Master Mix (New England Biolabs, Ipswich, MA, USA), 2 µM of forward and reverse primers, and about 10 ng template DNA. The thermal cycling program was set as follows: initial denaturation at 98 °C for 1 min, followed by 30 cycles of denaturation at 98 °C for 10 s, primer annealing at 50 °C for 30 s, and extension at 72 °C for 30 s, and a final extension at 72 °C for 5 min. For each leaf sample, the PCR was conducted in triplicate and the PCR products were pooled to make one PCR mixture. The mixture of PCR products was viewed by electrophoresis on 2% agarose gel, and a target DNA band was selected and purified with a Qiagen Gel Extraction Kit (Qiagen, Hilden, Germany). The library for amplicon sequencing was generated using TruSeq^®®^ DNA PCR-Free Sample Preparation Kit (Illumina, San Diego, CA, USA) and then sequenced on an Illumina NovaSeq platform (Novogene Bioinformatics Technology Co., Ltd., Tianjing, China).

After sequencing, the raw reads were filtered by Trimmomatic version 0.33 [20], and then their primer sequences were identified and removed by cutadapt version 1.9.1 [21]. High-quality reads were assembled by FLASH (version 1.2.7, http://ccb.jhu.edu/software/FLASH/ (accessed on 25 April 2022)). De-noising and removal of chimeric sequences were performed by dada2 [22] in QIIME2 2020.6 [23]. The amplicon sequence variants (ASVs) were clustered by dada2 and annotated by a Bayesian classifier using UNITE v7.2 (https://unite.ut.ee/ (accessed on 25 April 2022)) as a reference database. In addition, ASVs were assigned to different life forms by submitting them to the FUNGuild database (http://www.funguild.org/ (accessed on 25 April 2022)).

### 2.4. Statistical Analysis

The ASV table for all samples was generated in QIIME2 2020.6 [23]; the alpha diversity indices (observed species, Shannon, Simpson, ACE, Chao1, Good’s coverage, and phylogenetic diversity) of each sample and the relative abundance of each taxon were subsequently calculated in the same software. All statistical analyses were conducted in R v4.0.0 software [24]. Four commonly used statistical algorithms, including binary Jaccard, Bray–Curtis, weighted Unifrac, and unweighted Unifrac, were used to calculate the distances of the ASV matrices between samples in the vegan package [25]. PERMANOVA was carried out to assess the effects of orchards (Chengjiang and Xiyang) and sampling time (13 rounds of sampling) on the composition and structure of fungal communities using the adonis command in the vegan package. For visualization, the matrix of Bray–Curtis distance was subjected to principal coordinate analysis (PCoA) using the pcoa command in the Ape package [26]; the results were plotted using the package ggplot2 [27]. Shared ASVs between the old leaves and the spring leaves were defined using the packages FEAST [28] and VennDiagram [29], and then the Wilcoxon test was used to compare the average relative abundances of old leaves’ shared ASVs (Shared) and spring leaves’ unique ASVs (Unique). The regression analysis was carried out with a polynomial formula (y = x + I(x^2^)). The correlation analysis was carried out in the package Hmisc (https://hbiostat.org/R/Hmisc/ (accessed on 9 May 2022)). The data presented in a heatmap were analyzed and plotted in the package pheatmap [30]. Adjusted *p*-values (*fdr* adjusted) < 0.05 were accepted as significant.

## 3. Results

### 3.1. Temporal Succession of Leaf Fungal Communities

From March to October of 2021, we sampled the old leaves once (in mid-March, 12 samples, and 6 in each orchard) and the new leaves 12 times (144 samples). In total, we collected 156 leaf samples from the two orchards for sequencing. From all samples, 10,424,177 (mean number ± standard deviation, 66,822 ± 10,894) clean reads and 11,832 fungal ASVs were harvested (Appendix A). Then, 417 ASVs (with an average relative abundance ≥ 0.01%) were defined as core ASVs and used for further analysis. From the sequencing results, sampling time (*p* < 0.01) and orchard (*p* < 0.01) significantly affected the composition and structure of leaf fungal communities (Figure 1A), the samples from different sampling times scattered along principal coordinate 1 (PCoA 1, 22.7%), and those from Chengjiang and Xiyang scattered along PCoA 2 (14.7%). In both orchards, the temporal succession of fungal communities was typically represented by a decrease in the relative abundance of *Cladosporiaceae* (Figure 1B,C). However, the families that increased during the fungal succession were different in Chengjiang and Xiyang (Figure 1B,C). Specifically, the relative abundance of *Cladosporiaceae* decreased from 89.9 ± 1.3% at the end of March to 18.5 ± 15.5% at the end of August in Chengjiang (Figure 1B), and from 69.2 ± 9.5% to 18.3 ± 8.7% in Xiyang (Figure 1C). Besides *Cladosporiaceae*, the dominant families at the end of August in Chengjiang included *Capnodiales* (families incertae sedis, 15.7 ± 18.8%), *Trichomeriaceae* (10.4 ± 14.8%), and *Mycosphaerellaceae* (7.2 ± 6.7%), while in Xiyang, they included *Aureobasidiaceae* (11.8 ± 11.3%) and *Symmetrosporaceae* (7.5 ± 15%). In addition, the relative abundances of ASVs assigned to saprotrophs decreased in both orchards, the ones assigned to plant pathogens increased from 6.7 ± 6.3% at the end of March to 63.2 ± 16.4% at the end of August in Chengjiang (Figure 1D), and from 5.6 ± 5.1% to 80.5 ± 24.7% in Xiyang (Figure 1E).

### 3.2. Old Leaves’ Shared ASVs during Fungal Succession

Due to lateral transmission, the old leaves constitute a major fungal source for spring leaves. We calculated the ratios of relative fungal abundance in spring leaves that could be traced back to the old leaves and found that the ratios in both orchards generally decreased in the first several months, and then steadily varied between 20% and 60% (Figure 2A). Similarly, we calculated the distances of ASV matrices (represented by Bray–Curtis distance) between spring leaves and old leaves, and found that the distances to old leaves increased from the end of March to mid-August from 0.22 ± 0.05 to 0.77 ± 0.09 in Chengjiang (Figure 2B, *R^2^* = 0.71, *p* < 0.001), and from 0.28 ± 0.08 to 0.71 ± 0.18 in Xiyang (Figure 2C, *R^2^* = 0.67, *p* < 0.001). For the core ASVs, 81 of 292 (27.7%) and 68 of 342 (19.9%) ASVs of the spring leaves were shared with the old leaves in Chengjiang (Figure 2D) and Xiyang (Figure 2E), respectively. During the first several months, the average relative abundances of old leaves’ shared ASVs were significantly higher than those of the unique ASVs of spring leaves (Figure 2F,G). For example, in mid-June, the average relative abundances of old leaves’ shared ASVs were 0.76 ± 1.7% and 1.1 ± 2.7% compared to 0.39 ± 1.1% and 0.96 ± 3.6% of unique ASVs of spring leaves (both *p* < 0.0001) in Chengjiang and Xiyang, respectively. However, this advantage of shared ASVs was converted after mid-June; the average relative abundances of unique ASVs of spring leaves were consistently higher than their counterparts of old leaves’ shared ASVs (all *p* < 0.01).

### 3.3. Diseases and Climatic Factors Contributed to Fungal Succession

In the last two parts, we found that the fungal communities of spring leaves showed significant temporal variations at least from the end of March to mid-August (Figure 1A and Figure 2B,C). Besides community composition and structure, we found that the alpha diversity indices, represented by observed species, Shannon index, and phylogenetic diversity, also significantly altered during this period (Figure 3). When regressed to sampling time, the number of observed species and the Shannon and phylogenetic diversity indices showed a similar pattern to the variation in the ASV distance matrix (Figure 3). Specifically, the distribution of the number of observed species (*R^2^* = 0.68, *p* < 0.001 in Chengjiang; *R^2^* = 0.6, *p* < 0.001 in Xiyang; Figure 3A,D), Shannon index (*R^2^* = 0.73, *p* < 0.001; *R^2^* = 0.21, *p* < 0.001; Figure 3B,E), and phylogenetic diversity (*R^2^* = 0.69, *p* < 0.001; *R^2^* = 0.62, *p* < 0.001; Figure 3C,F) along sampling times all conformed to a curve of polynomial regression. In each curve, an increase stage was clearly defined from the end of March to mid-August, which was then followed by a decrease stage after mid-August.

To further understand the fungal temporal succession on spring leaves, we associated the alpha diversity indices (observed species, Shannon, phylogenetic diversity), principal coordinates (first three PCoAs calculated based on Bray–Curtis distance), and the distances of ASV matrices of spring leaves to old leaves (DM based on binary Jaccard, Bray–Curtis, weighted unifrac, and unweighted unifrac distances) with plant diseases and climatic factors (Figure 4). During the whole sampling period, we found the occurrence of citrus greasy spot and melanose in both orchards. Citrus greasy spot showed a significant negative correlation with fungal succession in Chengjiang (e.g., observed species *r* = −0.41, *p* < 0.001; Bray–Curtis distance *r* = −0.41, *p* < 0.001) and Xiyang (e.g., observed species *r* = −0.32, *p* < 0.01; Bray–Curtis distance *r* = −0.34, *p* < 0.01), while citrus melanose showed a significant positive correlation with fungal succession in Chengjiang (e.g., observed species *r* = 0.53, *p* < 0.001; Bray–Curtis distance *r* = 0.59, *p* < 0.001) and Xiyang (e.g., observed species *r* = 0.43, *p* < 0.001; Bray–Curtis distance *r* = 0.38, *p* < 0.001). The climatic factors, including precipitation and temperature, showed significant correlations with fungal succession. In comparison to both plant diseases and precipitation, the temperature factors (monthly average temperature, days of temperature > 35 °C, minimum and maximum temperature) showed the strongest correlation with fungal succession in Chengjiang (e.g., observed species *r* = 0.74, *p* < 0.001; Bray–Curtis distance *r* = 0.73, *p* < 0.001) and Xiyang (e.g., observed species *r* = 0.74, *p* < 0.001; Bray–Curtis distance *r* = 0.68, *p* < 0.001).

### 3.4. Dynamics of Fungal Families and Guilds Associated with Diseases and Climatic Factors

Finally, we found that plant diseases and climatic factors commonly showed positive effects on the top 50 fungal families on spring leaves. Specifically, these factors positively correlated with a clade consisting of 42 (84%) and 34 (68%) fungal families in Chengjiang and Xiyang, respectively (Figure 5). Among these factors, temperature had the strongest positive and consistent effects in the two orchards (Figure 5A,B) on different families including *Saccotheciaceae* (*r* = 0.5, *p* < 0.001 in Chengjiang; *r* = 0.4, *p* < 0.001 in Xiyang), *Erythrobasidiaceae* (*r* = 0.37, *p* < 0.001; *r* = 0.31, *p* < 0.01), *Xylariaceae* (*r* = 0.36, *p* < 0.01; *r* = 0.27, *p* = 0.02), *Periconiaceae* (*r* = 0.34, *p* < 0.01; *r* = 0.23, *p* = 0.04), *Ustilaginaceae* (*r* = 0.3, *p* < 0.01; *r* = 0.25, *p* = 0.03), *Trichomeriaceae* (*r* = 0.25, *p* = 0.03; *r* = 0.3, *p* < 0.01), and unclassified *Microbotryomycetes* (*r* = 0.24, *p* = 0.04; *r* = 0.33, *p* < 0.01); conversely, it had negative and consistent effects on different families including *Cladosporiaceae* (*r* = −0.65, *p* < 0.001; *r* = −0.52, *p* < 0.001), *Filobasidiaceae* (*r* = −0.28, *p* = 0.02; *r* = −0.58, *p* < 0.001), and *Debaryomycetaceae* (*r* = −0.26, *p* = 0.02; *r* = −0.57, *p* < 0.001). Precipitation showed positive and consistent effects on *Schizophyllaceae* (*r* = 0.52, *p* < 0.001; *r* = 0.28, *p* = 0.01) and *Symmetrosporaceae* (*r* = 0.41, *p* < 0.001; *r* = 0.38, *p* < 0.001). Citrus greasy spot (*r* = 0.48, *p* < 0.001; *r* = 0.35, *p* < 0.01) and melanose (*r* = −0.43, *p* < 0.001; *r* = −0.22, *p* = 0.05) showed opposite effects to *Cladosporiaceae* (Figure 5A,B). 

In addition, temperature positively correlated with the fungal guild of plant pathogens (*r* = 0.58, *p* < 0.001; *r* = 0.33, *p* < 0.01) and negatively correlated with undefined saprotrophs (*r* = −0.51, *p* < 0.001; *r* = −0.66, *p* < 0.001) in two orchards (Figure 5C,D). Precipitation positively correlated with litter saprotrophs (*r* = 0.48, *p* < 0.001; *r* = 0.48, *p* < 0.001), wood saprotrophs (*r* = 0.31, *p* < 0.01; *r* = 0.4, *p* < 0.001), fungal parasites (*r* = 0.3, *p* < 0.01; *r* = 0.48, *p* < 0.001), and plant saprotrophs (*r* = 0.23, *p* < 0.05; *r* = 0.39, *p* < 0.001). The outbreak of citrus melanose (*r* = 0.67, *p* < 0.001; *r* = 0.32, *p* < 0.01), but not citrus greasy spot, coincided with an increase in plant pathogens (Figure 5C,D).

## 4. Discussion

### 4.1. Fungal Temporal Succession Reduces the Priority Effects

Compared to annual and biennial plants, and also to the rhizosphere microbiome, the assembly and succession of the phyllosphere microbiome of perennial plants has received less attention and remained largely unknown [31,32]. The plant roots are known to produce highly diverse exudates to modulate the carbon, oxygen, pH, and nutrient availability of the rhizosphere, thus affecting the growth and activity of different microbial groups [33]. In contrast, the plant leaves leach out sugars and other nutrients for the survival of microbial communities [34]. The leaching process is highly associated with leaf wetness, and the concentration of leached carbohydrates is very low [34]. However, our study still observed a boom of fungal communities, which could be represented by the increases in observed species, Shannon index, and phylogenetic diversity from spring to summer (Figure 3). Even though some studies indicate that most leaf microbial species are introduced stochastically and are shaped by random processes [35,36], we found a consistent pattern of fungal temporal succession in both orchards (Figure 1). The temporal variation in all 417 core ASVs was largely represented by no more than ten fungal families (e.g., *Cladosporiaceae*, *Trichomeriaceae*, *Mycosphaerellaceae*, *Aureobasidiaceae*, and *Symmetrosporaceae*), and was also largely represented by the expansion of pathogen-related species. These results conform to the simplicity rule in microbial community assembly and succession [37,38].

Different from the seed-germinated plants, the assembled microbes of which are mainly from the planting soils [33], the emerged leaves obtained an important source of microbes from the older plant organs (Figure 2). Based on the analysis using a source-tracking algorithm [28], we inferred that the ASVs originating from the old leaves occupied the compartments of spring leaves immediately after emergence through horizontal transmission [33]. During the growth of spring leaves, the ASVs tracked to the old leaves decreased and were maintained at a stable level of 20% to 60% of the whole community. Instead, the unique ASVs and the pathogen-related species increased. This phenomenon may be explained by a reduction in the priority effects, in which the constructed community is challenged by more competitive fungal species and high environmental variability [8].

Our study did not differentiate leaf endophytes from epiphytes; the fungal endophytes may transmit from old leaves to new leaves, thus occupying the endosphere niche of the new leaves for a short time. Different to the epiphytes, fungal endophytes usually construct a stable relationship with their host plant, so their occurrences are not stochastic [39]. This may be one of the reasons why a stable level of fungal ASVs in spring leaves could be traced back to the old leaves.

### 4.2. Fungal Succession with Boosted Fungal Groups Related to Citrus Pathogens

In comparison to other microbial species, plant pathogens have been shown to have some advantages in colonizing plant organs and to be key players in altering the composition and structure of plant microbial communities [39,40,41]. During the fungal succession of the spring leaves, the relative abundances of the ASVs assigned to potential pathogens increased to over 60% after the decrease in the ASVs tracked to the old leaves (Figure 1D,E). These results imply that the reduction in priority effects in the microbial communities of spring leaves may correlate to the spread of plant pathogens. For example, the relative abundance of *Mycosphaerellaceae*, hosting the causal agent of citrus greasy spot, increased at the end of August compared to the end of March (Figure 1B,C). Additionally, the increased *Mycosphaerellaceae* correlated with the outbreak of citrus greasy spot in both orchards (Figure 5A,B), so we infer that the increased *Mycosphaerellaceae* was partly caused by an increase in the causal agent of citrus greasy spot.

In addition, the relative abundance of *Diaporthaceae*, hosting the causal agent of citrus melanose, was extremely low (<0.01% in both orchards) and was not included in the core families. This may be explained by the fact that the occurrence of citrus greasy spot, not melanose, relies on a stage of endophytic growth in citrus leaves [15,16]. Additionally, the field experiment was human-disturbed; the application of some fungicides (about twice a month, Appendix A), targeted to the causal agent of citrus melanose, might have caused the low presence of *Diaporthaceae* on the leaves. Except for the causal agents of citrus greasy spot and melanose, other pathogen-related fungal groups (e.g., *Colletotrichum*, *Lasiodiplodia*, *Phyllosticta* groups) were not observed abundantly in our sequencing data, which confirms that their caused citrus diseases have not been recorded in these orchards for many years [2,3,4].

From the above, we infer that the ASVs of *Mycosphaerellaceae* may contribute to boosting the citrus pathobiota during the fungal succession of spring leaves (Figure 1D,E), during which some potential pathogens transform to an endophytic lifestyle to facilitate their succession [42]. However, we only analyzed and predicted the citrus pathogen-associated species at the family level (e.g., *Diaporthaceae* and *Mycosphaerellaceae*); the actual abundances of citrus pathogens still need to be quantitatively detected to confirm these results.

### 4.3. Seasonal Temperature and Precipitation Variations Favor Fungal Spread and Affect Fungal Succession

It is well known that the variation in climatic factors affects the growth and spread of many plant fungal pathogens, especially the increase in temperature and precipitation from spring to summer [14,15,16]. Similarly, our study confirms that the seasonal variation in temperature and precipitation also contributed to the boom of other leaf fungal species (Figure 3 and Figure 4); both the observed species and species diversity were increased relative to the increase in temperature and precipitation. On the contrary, we observed a slight decrease in the observed species and species diversity after the end of August, which may be caused by the decrease in the temperature and precipitation from summer to autumn. In sum, we infer that climatic factors affect the fungal succession of spring leaves directly and indirectly by jointly working with plant pathogens.

## 5. Conclusions

For perennial pomelo plants, the assembly and succession pattern of fungal communities on spring leaves is typically represented by (1) the reduction in priority effects given by the fungal species from old leaves; (2) the decrease in the relative abundance of *Cladosporiaceae*, followed by the occurrence of more diverse fungal families; and (3) the increase in the number of observed species, Shannon and phylogenetic diversity indices, and pathogen-associated species. In addition, this pattern is jointly affected by disease occurrence and climatic factors, i.e., the succession of phyllosphere fungal communities is highly correlated to the temperature increase from spring to summer.

## Figures and Tables

**Figure 1 microorganisms-12-01157-f001:**
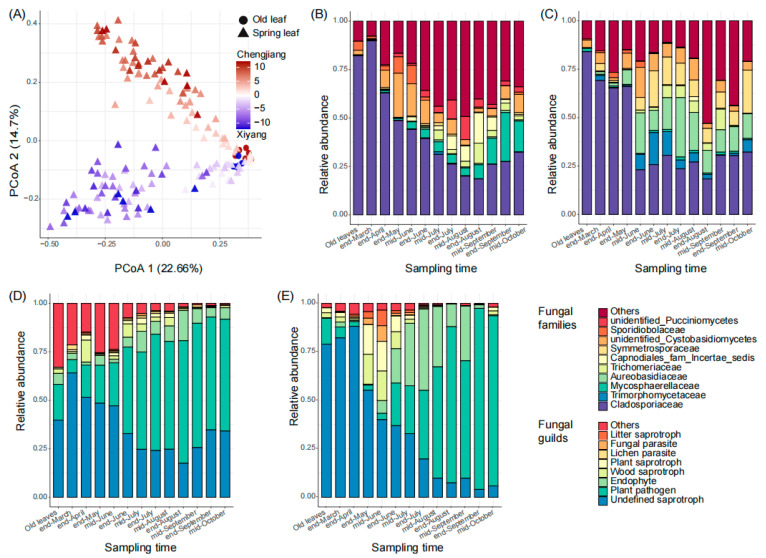
Temporal succession of spring leaf fungal communities in Chengjiang and Xiyang. (**A**) Plot of the results of the principal coordinate analysis of leaf samples based on Bray–Curtis distance of the ASV matrix. (**B**,**C**) Temporal dynamics of spring leaves’ top 10 fungal families from the end of March to mid-October in Chengjiang (**B**) and Xiyang (**C**). (**D**,**E**) Temporal dynamics of spring leaf fungal guilds from the end of March to mid-October in Chengjiang (**D**) and Xiyang (**E**).

**Figure 2 microorganisms-12-01157-f002:**
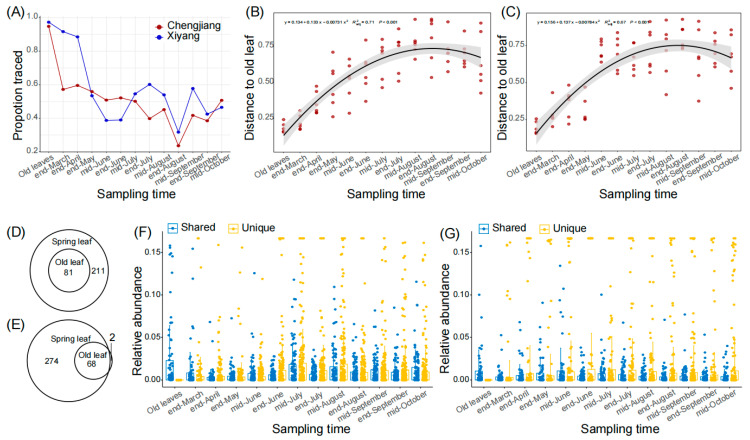
Old leaves’ shared ASVs during spring leaf fungal succession in Chengjiang and Xiyang. (**A**) Proportions of relative fungal abundances of spring leaves that were traced to old leaves from the end of March to mid-October. (**B**,**C**) Bray–Curtis distances of ASV matrices between spring leaves and old leaves from the end of March to mid-October in Chengjiang (**B**) and Xiyang (**C**). (**D**,**E**) Old leaves and spring leaves shared core ASVs in Chengjiang (**D**) and Xiyang (**E**). (**F**,**G**) Comparison of the average relative abundances of old leaves’ shared ASVs (Shared, blue) and spring leaves’ unique ASVs (Unique, yellow) from the end of March to mid-October in Chengjiang (**F**) and Xiyang (**G**).

**Figure 3 microorganisms-12-01157-f003:**
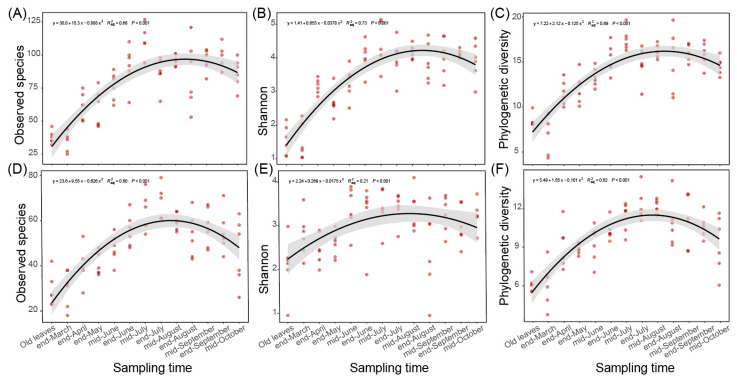
Variation in the alpha diversity indices during spring leaf fungal succession. (**A**–**C**) The number of observed species (**A**), Shannon index (**B**), and phylogenetic diversity (**C**) regressed with sampling time in Chengjiang. (**D**–**F**) The number of observed species (**D**), Shannon index (**E**), and phylogenetic diversity (**F**) regressed with sampling time in Xiyang.

**Figure 4 microorganisms-12-01157-f004:**
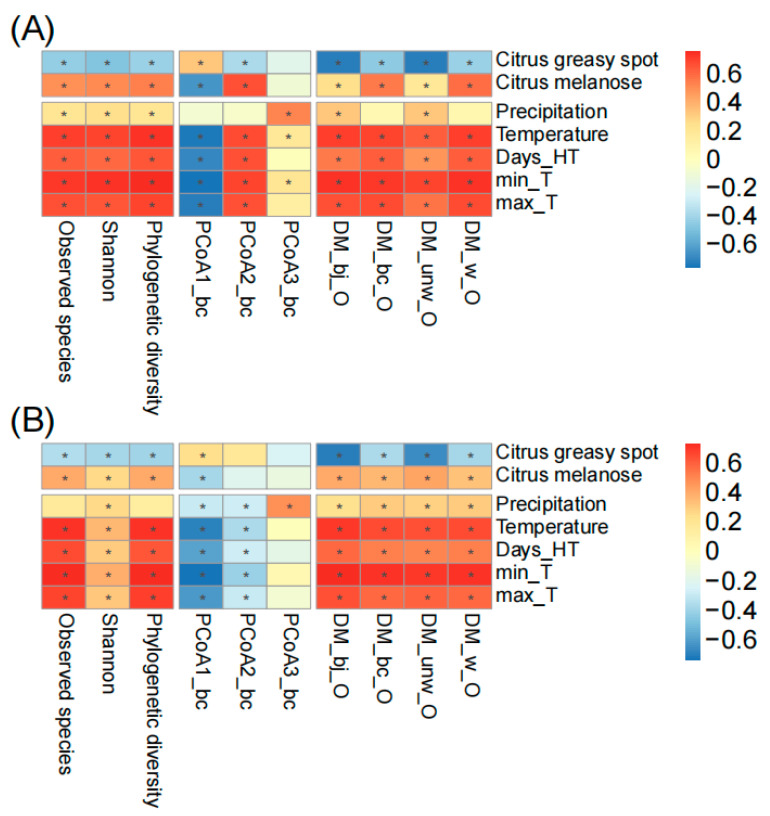
The correlation analysis of plant diseases and climatic factors with spring leaf fungal succession in Chengjiang (**A**) and Xiyang (**B**). The correlation coefficients are presented as heatmap plots. Fungal succession is represented by alpha diversity indices (observed species, Shannon index, phylogenetic diversity), principal coordinates (first three PCoAs calculated based on Bray–Curtis distance), and the distances of ASV matrices of spring leaves to old leaves (DM based on binary Jaccard, Bray–Curtis, weighted unifrac, and unweighted unifrac distances). Days_HT, monthly number of days of temperature > 35 °C; min_T and max_T, monthly minimum and maximum temperature. The adjusted *p*-values ≤ 0.05 are marked with “*”.

**Figure 5 microorganisms-12-01157-f005:**
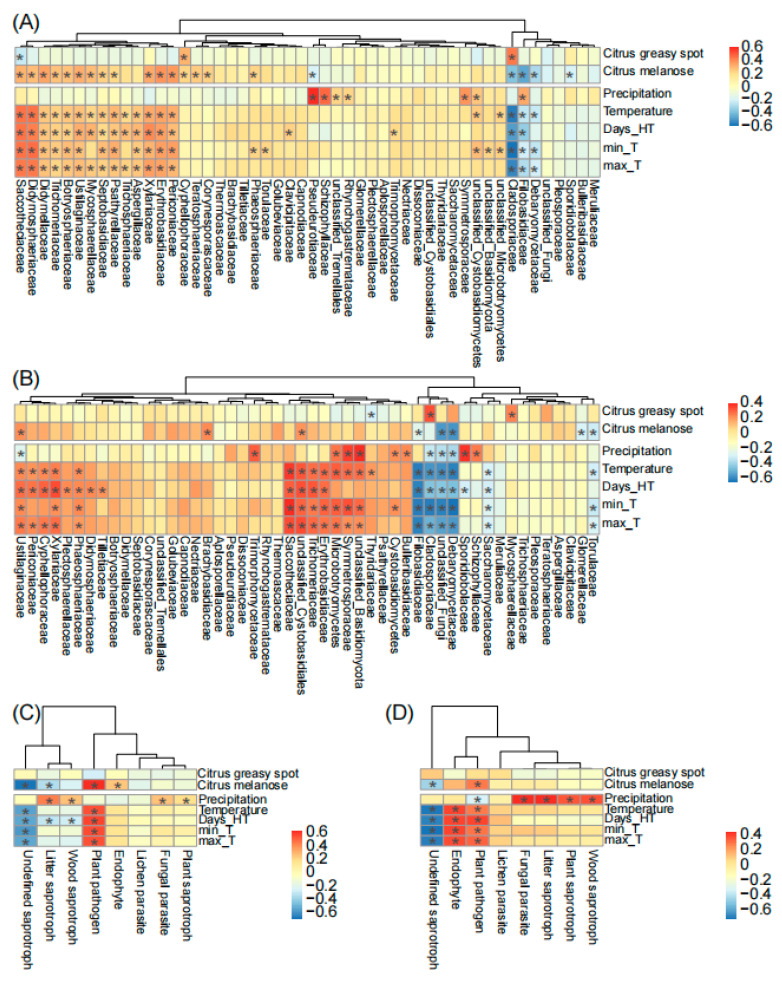
The correlation analysis of plant diseases and climatic factors with fungal families (**A**,**B**) and fungal guilds (**C**,**D**) during spring leaf fungal succession in Chengjiang and Xiyang. Days_HT, monthly number of days of temperature > 35 °C; min_T and max_T, monthly minimum and maximum temperature. The adjusted *p*-values ≤ 0.05 are marked with “*”.

## Data Availability

Raw sequence data for ITS were deposited under NCBI BioProject Accession No. PRJNA1115902. Data are contained within the article and Appendix A. The codes in analyses and plots are available from the corresponding author on reasonable request.

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
