# Peer review of "Disease Occurrence and Climatic Factors Jointly Structure Pomelo Leaf Fungal Succession in Disturbed Agricultural Ecosystem"

_microorganisms, 2024, doi:10.3390/microorganisms12061157_

Round 1

Reviewer 1 Report

Comments and Suggestions for Authors

An interesting study that explores the succession of saprophytic and pathogenic fungi on young leaves in two pomelo orchards over time as they mature in comparison with the previous seasons old leaves.

The Introduction provides suitable background to the project. The last paragraph, however, is poorly constructed. Instead of giving some of the study’s findings, it is more appropriate to explain how you executed the study to test the hypothesis.

In the methods: add more information on the orchard size and lie of the land; did rainfall events occur during the sampling season?; what is the meaning of “blocks were lined out”?; and, for the pomelo disease assessment, were any published guides used for this? What weather stations were used in the study and what were their proximities to the orchards that were sampled?

The results clearly illustrate that on new leaves, the saprotrophs decreased and the pathogens increased substantially with time but there was a strong site effect in their makeup. The relative importance of the fungal domain of old leaves is clearly explained through the analysis of ASVs. The diversity analysis and climate correlations provide insight into the spatial and temporal dynamics of the foliar fungal families and guilds.

The figures are excellent but in 1 and 5 the font is rather small and therefore difficult to read. If space permits, consider enlarging the figures so the font size is similar to that in Figure 4.

Since whole leaves were ground before DNA extraction, the Discussion should include mention of the likely contribution of endophytes to the overall findings. Please also discuss the extent to which fungicides were used in the orchards and whether there had been any previous management of leaf spots and other pathogens.

The inclusion of a small section titled Conclusions is suggested.

 I recommend replacing “sprouted” with “emerged” throughout as the former generally refers to the whole fresh shoots and this study only addresses the leaves.

Comments on the Quality of English Language

The English needs extensive editing. The errors range from incomplete sentences (lines 14-15) to spelling (epidermics in line 75) to poor syntax.

Author Response

Comments and Suggestions for Authors

An interesting study that explores the succession of saprophytic and pathogenic fungi on young leaves in two pomelo orchards over time as they mature in comparison with the previous seasons old leaves.

Response: Thank you for this comment on our manuscript. This will largely encourage us to continue on this kind of projects.

The Introduction provides suitable background to the project. The last paragraph, however, is poorly constructed. Instead of giving some of the study’s findings, it is more appropriate to explain how you executed the study to test the hypothesis.

Response: Done as suggested. We reorganized the last paragraph of the Introduction, please check it again.

In the methods: add more information on the orchard size and lie of the land; did rainfall events occur during the sampling season?; what is the meaning of “blocks were lined out”?; and, for the pomelo disease assessment, were any published guides used for this? What weather stations were used in the study and what were their proximities to the orchards that were sampled?

Response: We added more information on this part. The rainfall events were often seen during our sampling season. We tried our best to avoid the raining days, but the rain was sometimes very heavy and lasted for days, so we didn’t finish the investigation and sampling for several times.

The “lined out” was substituted by “set”, I hope it’s clearer this time.

For pomelo disease assessment, we did not find any guides. I used one of the methods from my textbook “Phytopathology” (in Chinese), it was a textbook we used at the University.

The used weather station was managed by the Bureau of Meizhou Meteorlogical Service,they only offers average temperature and precipitation data of the whole district. We are sorry for we couldn’t offer the specific data for each orchard.

The results clearly illustrate that on new leaves, the saprotrophs decreased and the pathogens increased substantially with time but there was a strong site effect in their makeup. The relative importance of the fungal domain of old leaves is clearly explained through the analysis of ASVs. The diversity analysis and climate correlations provide insight into the spatial and temporal dynamics of the foliar fungal families and guilds.

Response: The reviewer is right on this. The neglect of site effect was one drawback of our experimental design, we should have considered this. As we know, the citrus fungal pathogens overwinter on the leave and twig litters in the orchard, and during the spring and summer days, they germinate and sporulate to disseminate their spores and conidia to the surface of citrus leaves. This might cause that the pathogens increased substantially along with our sampling time. Hope the review think this is reasonable.

It’s really appreciated for the positive comments on the Results.

The figures are excellent but in 1 and 5 the font is rather small and therefore difficult to read. If space permits, consider enlarging the figures so the font size is similar to that in Figure 4.

Response: Thank you for this suggestion. We found some figures were hard to read, and adjusted the font size. Hope the figures are better to read this time.

Since whole leaves were ground before DNA extraction, the Discussion should include mention of the likely contribution of endophytes to the overall findings. Please also discuss the extent to which fungicides were used in the orchards and whether there had been any previous management of leaf spots and other pathogens.

Response: Thank you for these suggestions. The endophytes were discussed in 4.1 and 4.2 of the Discussion. In 4.1, we discussed the endophytes might deter the reduce of priority effects. And in 4.2, we discussed that some pathogens might live as endophytes in citrus before their infection. Please check it again.

About the fungicides, we added a table S1, and added some sentences in 4.2 of Discussion.

The inclusion of a small section titled Conclusions is suggested.

Response: Done as suggested. A Conclusion part is added.

I recommend replacing “sprouted” with “emerged” throughout as the former generally refers to the whole fresh shoots and this study only addresses the leaves.

Response: Done as suggested.

Comments on the Quality of English Language

The English needs extensive editing. The errors range from incomplete sentences (lines 14-15) to spelling (epidermics in line 75) to poor syntax.

Response: We asked for help from one of my old friends when I was at The University of Chicago, I hope it reads better this time.

Reviewer 2 Report

Comments and Suggestions for Authors

Comments to authors

Disease occurrence and climatic factor jointly structure leaf fungal succession in disturbed agricultural ecosystem

The paper is valuable contribution to knowledge of the fungal community and fungal succession pattern in pomelo orchards. These results of temporal fungal succession in community can add to our understanding of the epidemiology of plant pathogens.

Title:

Add name of plant into the title e.g. pomelo or citrus

Introduction:

The introduction can benefit from a clearer flow of ideas to guide the reader through the various aspects of the topic, also some sentences can be simplified for clarity and conciseness, and related ideas can be consolidated to avoid repetition.

Common comments - clarify the flow:, simplify and consolidate and ensure consistent style, e.g. maintain consistency in referencing studies and concepts throughout the text

Line 38-46 – this section lacks clarity regarding the specific control methods employed for diseases attributed to the Phyllosticta genus and the Elsinoë genus. Furthermore, it does not address whether fungicide treatments may have influenced changes in the microbial community. It is imperative to elucidate the strategies implemented for disease control and assess potential factors such as fungicide resistance or climatic conditions that may have impacted the prevalence of these significant diseases  

Line 68: please also add data on mycorrhizal fungi. As far as I know there are a at least few study focused on evaluation evaluated different citrus varieties' responses to arbuscular mycorrhizal fungi and different pathogen, e,g, Phytophthora parasitica. Also it should be in Discussion – e.g. inoculation with arbuscular mycorrhizal fungi might reduce disease severity across all citrus plants, including pomelo.

.

Line 70-79: please add aim or/and objectives after hypothesis.

M&M

Lines 98-99: how disease investigation has been done, just check the visual symptoms; how diseases severity has been evaluated, how diseases were identified? Please add and clarify part of disease investigation.

Line 121 – please add reference/s for primers, it seems to me that ITS 1R is ITS2 primer (5′-GCTGCGTTCTTCATCGATGC-3′).

Line 149 – add  R packages and reference for statistical analysis

Results and Discussion

I couldn't find in the results how you confirmed Citrus black spot, caused by Phyllosticta species, citrus scab, caused by Elsinoë species, citrus melanose, caused by Diaporthe species, and citrus greasy spot, caused by Zasmidium species for each leaf. Please consider adding relevant data either in the Methods or Results section. Additionally, your study primarily focuses on the temporal succession of spring leaf fungal communities, but your hypothesis emphasizes fungal pathogens. Could you provide clarification on this discrepancy?

Please add conclusions.

Comments on the Quality of English Language

Minor editing of English language required

Author Response

Comments and Suggestions for Authors

Comments to authors

Disease occurrence and climatic factor jointly structure leaf fungal succession in disturbed agricultural ecosystem

The paper is valuable contribution to knowledge of the fungal community and fungal succession pattern in pomelo orchards. These results of temporal fungal succession in community can add to our understanding of the epidemiology of plant pathogens.

Response: Thank you for the comments on our manuscript.

Title:

Add name of plant into the title e.g. pomelo or citrus

Response: Done as suggested. We added pomelo as it’s more specific.

Introduction:

The introduction can benefit from a clearer flow of ideas to guide the reader through the various aspects of the topic, also some sentences can be simplified for clarity and conciseness, and related ideas can be consolidated to avoid repetition.

Common comments - clarify the flow:, simplify and consolidate and ensure consistent style, e.g. maintain consistency in referencing studies and concepts throughout the text

Response: Done as suggested. We revised the Introduction to make it clearer and more concise, and reorganized the last paragraph. Please check it again.

Line 38-46 – this section lacks clarity regarding the specific control methods employed for diseases attributed to the Phyllosticta genus and the Elsinoë genus. Furthermore, it does not address whether fungicide treatments may have influenced changes in the microbial community. It is imperative to elucidate the strategies implemented for disease control and assess potential factors such as fungicide resistance or climatic conditions that may have impacted the prevalence of these significant diseases  

Response: The review is right, we did not arrange the sentences in causality. We added more sentences in this paragraph to make it reasonable. Please check it again.

Line 68: please also add data on mycorrhizal fungi. As far as I know there are a at least few study focused on evaluation evaluated different citrus varieties' responses to arbuscular mycorrhizal fungi and different pathogen, e,g, Phytophthora parasitica. Also it should be in Discussion – e.g. inoculation with arbuscular mycorrhizal fungi might reduce disease severity across all citrus plants, including pomelo.

Response: The reviewer is right, mycorrhizal fungi and Phytophthora parasitica are closely related to plant, but they are commonly found in plant roots. We looked through our data, and found the fungal species of Glomeromycota and Phytophthora were extremely low in relative abundance. Based on this, we did not include the results of Glomeromycota and Phytophthora in our manuscript. I hope the reviewer could understand us, we sincerely wanted to solve his concern.

Line 70-79: please add aim or/and objectives after hypothesis.

Response: This paragraph was reorganized, please check it again.

M&M

Lines 98-99: how disease investigation has been done, just check the visual symptoms; how diseases severity has been evaluated, how diseases were identified? Please add and clarify part of disease investigation.

Response: We largely revised the 2.1 part to make sure it’s easier to read. For citrus diseases like Huanglongbing, melanose and canker, we can use the molecular method like quantitative PCR to detect them. For some other diseases, we also can isolate the pathogen and identify them based on ITS sequences. In our investigation, we only saw the occurrence of greasy spot and melanose, they have outbroken in the orchards for several years. Their symptoms are very typical, and I have worked on them for 14 years, so we did not identify them by molecular methods.

Line 121 – please add reference/s for primers, it seems to me that ITS 1R is ITS2 primer (5′-GCTGCGTTCTTCATCGATGC-3′).

Response: Done as suggested. The ITS 1R is ITS2, we corrected it. Thank you for your reminding.

Line 149 – add  R packages and reference for statistical analysis

Response: Done as suggested.

Results and Discussion

I couldn't find in the results how you confirmed Citrus black spot, caused by Phyllosticta species, citrus scab, caused by Elsinoë species, citrus melanose, caused by Diaporthe species, and citrus greasy spot, caused by Zasmidium species for each leaf. Please consider adding relevant data either in the Methods or Results section. Additionally, your study primarily focuses on the temporal succession of spring leaf fungal communities, but your hypothesis emphasizes fungal pathogens. Could you provide clarification on this discrepancy?

Response: This is a very good question to our manuscript. For this, I have checked the DNA of our leaf samples, but found there are not enough DNA left for us to add the data. It’s a drawback of our experimental design, we should have included quantitative PCR for the pathogens. Based on these, we watched out our language more carefully this time. We use “pathogen associated species”, rather than “pathogens”, and explain it in 4.2.

Please add conclusions.

Response: Done as suggested.

Comments on the Quality of English Language

Minor editing of English language required

Response: The language was revised by one of my old friends when I was at The University of Chicago, I hope it reads better this time.